DOI: 10.1038/s41467-018-04090-2　　**OPEN**

# Regulation of Yki/Yap subcellular localization and Hpo signaling by a nuclear kinase PRP4K

Yong suk Cho[1], Jian Zhu[1,2], Shuangxi Li[1], Bing Wang[1], Yuhong Han[1] & Jin Jiang[1,3]

Hippo (Hpo) signaling pathway controls tissue growth by regulating the subcellular localization of Yorkie (Yki)/Yap via a cytoplasmic kinase cassette containing an upstream kinase Hpo/MST1/2 and a downstream kinase Warts (Wts)/Lats1/2. Here we show that PRP4K, a kinase involved in mRNA splicing, phosphorylates Yki/Yap in the nucleus to prevent its nuclear accumulation and restrict Hpo pathway target gene expression. PRP4K inactivation accelerates whereas excessive PRP4K inhibits Yki-driven tissue overgrowth. PRP4K phosphorylates a subset of Wts/Lats1/2 sites on Yki/Yap to inhibit the binding of Yki/Yap to the Scalloped (Sd)/TEAD transcription factor and exclude Yki/Yap nuclear localization depending on nuclear export. Furthermore, PRP4K inhibits proliferation and invasiveness of cultured breast cancer cells and its high expression correlates with good prognosis in breast cancer patients. Our study unravels an unanticipated layer of Hpo pathway regulation and suggests that PRP4K-mediated Yki/Yap phosphorylation in the nucleus provides a fail-safe mechanism to restrict aberrant pathway activation.

[1] Department of Molecular Biology, University of Texas Southwestern Medical Center at Dallas, Dallas, TX 75390, USA. [2] Henan Key Laboratory of immunology and targeted therapy, School of Laboratory Medicine, Henan Collaborative Innovation Center of Molecular Diagnosis and Laboratory Medicine, Xinxiang Medical University, Xinxiang, 453003 Henan Province, People's Republic of China. [3] Department of Pharmacology, University of Texas Southwestern Medical Center at Dallas, TX 75390 Dallas, USA. Correspondence and requests for materials should be addressed to J.J. (email: jin.jiang@utsouthwestern.edu)

How organs know when they reach final size and stop growth during development is an important yet still poorly understood problem in modern biology. The regulation of tissue growth and organ size depends on a delicate balance between cell proliferation and cell death, which is tightly controlled by both global and local cues. Indeed, tissue growth is not only influenced by environmental factors, such as hormonal signals and nutrients, but also by organ-intrinsic mechanisms. The Hippo (Hpo) signaling pathway has emerged as an evolutionarily conserved developmental pathway that controls tissue growth and organ size in species ranging from *Drosophila* to mammals[1–4], and deregulation of Hpo pathway activity has been implicated in many human cancers[4,5].

The core of the Hpo pathway is a conserved kinase cassette consisting of an upstream kinase Hpo/MST1/2, members of the Ste20 kinase family[6–10], and a downstream kinase Warts (Wts)/Lats1/2, members of the Nuclear Dbf-2-related (NDR) kinase family[11,12]. Hpo/MST1/2 phosphorylates and activates Wts/Lats1/2, which in turn phosphorylates and inactivates Yorkie (Yki)/YAP[13,14]. Recent studies indicate that MAP4K family members act in parallel with Hpo/MST1/2 to regulate Wts/Lats1/2 and Yki/YAP in both *Drosophila* and mammalian cells[15–18]. Phosphorylation of Yki/YAP restricts its nuclear localization in part through recruiting 14-3-3[19–22]. When the activity of the kinase cascade is compromised, unphosphorylated Yki/YAP enters the nucleus and interacts with the TEAD family transcription factors Scalloped (Sd) in *Drosophila* and TEAD1-4 in mammals[21,23–25], leading to activation of Hpo pathway target genes that regulate cell growth, proliferation, and survival.

How Yki/YAP is regulated in the nucleus is still poorly understood. Here, we conducted a kinome screen and identified PRP4K as a genetic modifier of the eye overgrowth phenotype caused by Yki overexpression. PRP4K is a nuclear kinase implicated in the regulation of mRNA splicing and spindle assembly checkpoint[26–29]; however, it has not been implicated in Hpo signaling and organ size control. We provided both genetic and biochemical evidence that PRP4K regulated Hpo signaling by phosphorylating Yki/Yap. We found that PRP4K acted in parallel with Wts/Lats1/2 to phosphorylate a subset of Wts/Lats1/2 sites on Yki/Yap, and that PRP4K-mediated phosphorylation restricted Yki/Yap activity by attenuating its binding to the Scalloped (Sd)/TEAD transcription factor and excluding its nuclear localization. Furthermore, PRP4K inhibited proliferation and invasiveness of cultured breast cancer cells and its high expression correlated with good prognosis in triple-negative breast cancer patients.

## Results

**Genetic screen identified PRP4K as a novel component in the Hpo pathway.** To identify additional Hpo pathway regulators, we conducted a genetic modifier screen in which we used transgenic RNAi to inactivate individual genes and determined whether knockdown of the targeted genes modified the tissue overgrowth phenotype caused by Yki overexpression in *Drosophila* eyes (*GMR>Yki*)[16,30]. By screening through a collection of transgenic RNAi lines targeting the *Drosophila* kinome, we identified several kinases including Hpo, aPKC, Tao-1, Misshapen (Msn), Happy hour (Hppy), and PRP4 kinase (PRP4K; also known as PRPF4B or PRP4) whose inactivation enhanced *GMR>Yki* induced eye overgrowth (Fig. 1a–c, e)[16]. To validate that PRP4K is involved in modifying the *GMR>Yki* phenotype, we employed three independent transgenic RNAi lines (*PRP4K^{i-1}* or *PRP4K^i* for simplicity, *PRP4K^{i-2}*, *PRP4K^{i-3}*) that target different regions of PRP4K coding sequence and found that they enhanced *GMR>Yki* induced eye overgrowth in a similar fashion, although expression

of these RNAi transgenes with *GMR* in otherwise wild-type eyes did not cause a discernible change in eye size (Supplementary Fig. 1a–j). In addition, PRP4K RNAi elevated the expression of a Hpo pathway target gene *diap1-GFP3.5* induced by *GMR>Yki* (Figs. 1a–c, 1f)[21]. On the other hand, PRP4K overexpression suppressed *diap1-GFP3.5* expression and eye overgrowth induced by *GMR>Yki* (Fig. 1d–f). In corroborating with the phenotypes caused by PRP4K RNAi, *GMR>Yki* eyes carrying clones homozygous for a *PRP4K* mutation (*PRP4K^{EY11156}*) grew larger than *GMR-Yki* eyes carrying control clones (Fig. 1g–k). In addition, *PRP4K* mutant clones (marked by orange color in Fig. 1h) occupied larger areas of the eyes compared with control clones (marked by white color in Fig. 1g), suggesting that *PRP4K* mutant cells have a growth advantage over wild-type cells.

PRP4K RNAi did not enhance eye overgrowth caused by overexpression of a constitutively active insulin receptor (*GMR>InR^{CA}*; Supplementary Fig. 1k-I, 1o), suggesting that PRP4K modulates tissue growth in a Hpo-pathway-specific manner. PRP4K RNAi also failed to enhance the eye overgrowth phenotype caused by expressing an Yki-independent and constitutively active form of Sd (Sd-GA; Fig. 1l, m, p)[21,23,24]. On the other hand, overexpression of PRP4K suppressed the eye overgrowth caused by inactivation of Wts (Fig. 1n, o, q). These results suggest that PRP4K controls tissue growth and organ size by selectively regulating the Hpo pathway downstream of or in parallel with Wts but upstream of Sd. The ability of PRP4K to regulate Hpo signaling depends on its kinase activity because overexpression of a kinase dead form of PRP4K (PRP4K^{KR}; K617R) failed to suppress the *GMR>Yki* phenotype (Supplementary Fig. 1n compared with 1m, 1p).

**PRP4K phosphorylates Yki to restrict its nuclear localization.** PRP4K RNAi in the posterior compartment of wing imaginal discs (*hh>PRP4K^i*) increased the expression of multiple Hpo pathway target genes including *expanded* (*ex-lacZ*), *diap1* (*diap1-GFP3.5*), *bantam* (*ban-lacZ*), and *wingless* (*wg*) (Fig. 2a–h; Supplementary Fig. 2a-d). Co-expression of PRP4K or knockdown of Yki suppressed the elevated *ex-lacZ* expression caused by *hh>PRP4K^i* whereas co-expression of PRP4K^{KR} failed to do so (Supplementary Fig. 2e-i). *hh>PRP4K^i* resulted in an increase in the level of nuclear Yki in posterior compartment cells (Fig. 2i, j). On the other hand, overexpression of PRP4K suppressed the nuclear accumulation of Yki in *wts* mutant clones (Fig. 2k, l), consistent with the observation that excessive PRP4K was able to suppress tissue overgrowth caused by Wts RNAi (Fig. 1n, o).

The observations that PRP4K inactivation resulted in increased Yki nuclear localization and that PRP4K acts downstream of or in parallel with Wts prompted us to determine whether PRP4K regulates Hpo signaling by phosphorylating Yki. Indeed, co-expression of PRP4K but not PRP4K^{KR} with Myc-Yki in S2 cells induced a mobility shift of Myc-Yki on a phospho-tag gel that reduces the mobility of phosphorylated proteins (Fig. 3a)[31]. PRP4K-induced mobility shift of Myc-Yki was eliminated by treating the cell extracts with a phosphatase (Supplementary Fig. 3a), confirming that the mobility shift was due to phosphorylation. Wts RNAi abolished Myc-Yki mobility shift induced by Hpo co-expression, but did not affect Myc-Yki mobility shift caused by PRP4K co-expression (Fig. 3a, b), suggesting that PRP4K promotes Yki phosphorylation independent of Wts. Furthermore, Yki phosphorylation was reduced in eye discs containing *PRP4K* mutant clones (Fig. 3c). Tethering Yki to the cytoplasm compartment by addition of a myristoylation signal (Myr-Yki) abolished PRP4K-mediated but not Hpo-mediated phosphorylation of Yki (Supplementary Fig. 3c compared with Supplementary Fig. 3b). On the other hand,

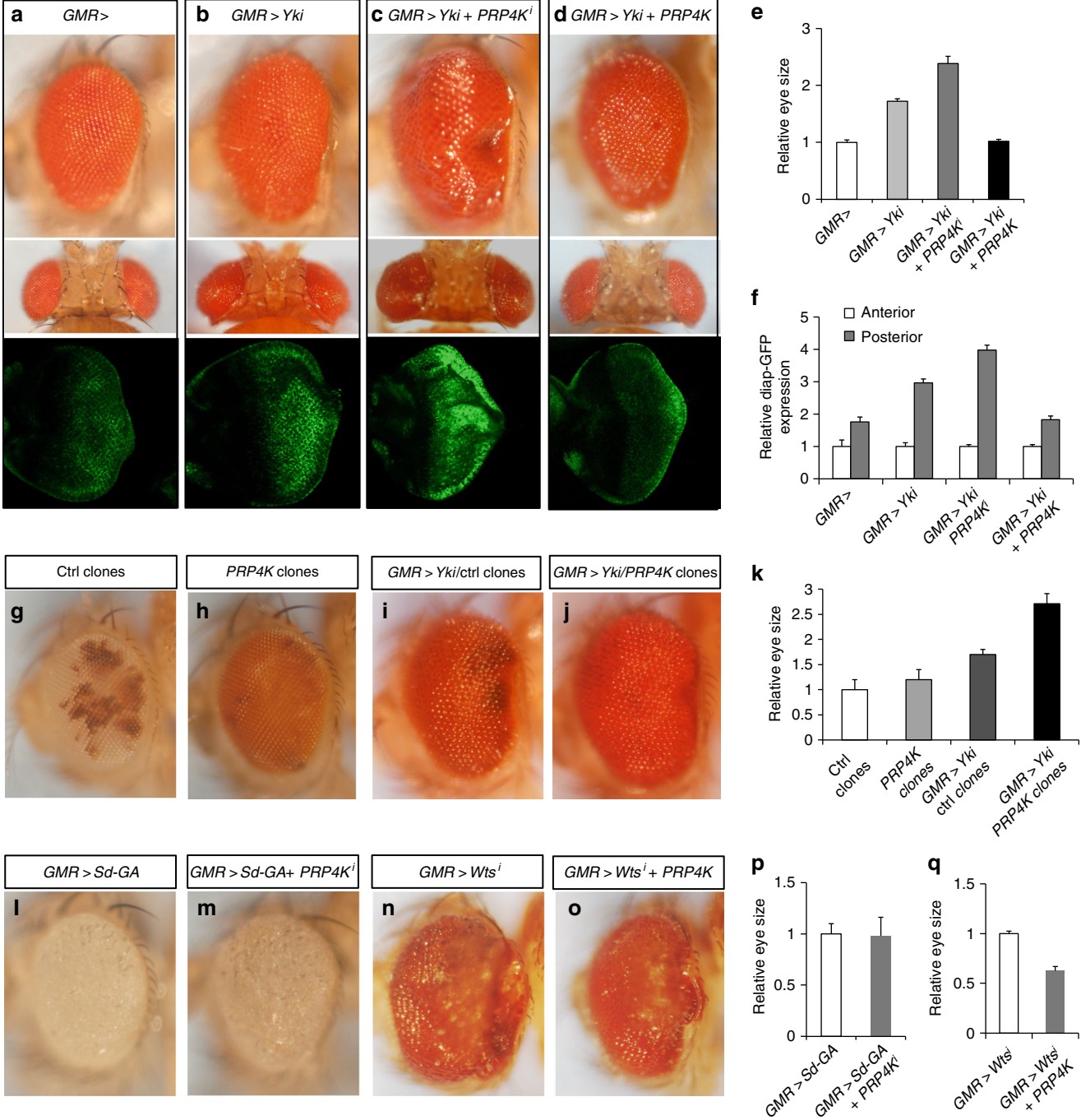

**Fig. 1** PRP4K regulates organ size through the Hpo pathway. **a–d** Side (top) or dorsal view (middle) of *Drosophila* adult eyes or late third instar eye imaginal discs expressing *diap-GFP3.5* (bottom) of the following genotypes: *GRM-Gal4* (**a**), *GMR-Gal4 UAS-Yki* (**b**), *GMR-Gal4 UAS-Yki UAS-PRP4K-RNAi* (**c**), and *GMR-Gal4 UAS-Yki UAS-PRP4K* (**d**). **e** Quantification of eye size for the indicated genotypes. **f** Quantification of GFP signal in the indicated eye discs in anterior and posterior regions relative to the morphogenetic furrow. **g–j** *Drosophila* adult eyes containing control clones without (**g**; white color) or with (**i**) *GMR>Yki*, or *PRP4K* mutant clones without (**h**; orange color) or with (**j**) *GMR>Yki*. (**k**) Quantification of eye size for the indicated genotypes. **l–m** *Drosophila* adult eyes expressing *UAS-Sd-GA* either alone (**l**) or in conjunction with *UAS-PRP4K-RNAi* (**m**) under the control of *GMR-Gal4*. **n–o** *Drosophila* adult eyes expressing *UAS-Wts-RNAi* either alone (**n**) or in conjunction with *UAS-PRP4K* (**o**) under the control of *GMR-Gal4*. **p–q** Quantification of eye size for the indicated genotypes. Data are means ± s.d. from three independent experiments. $N \geq 5$ for each genotype

inhibition of nuclear export by treating cells with LMB increased PRP4K-mediated phosphorylation but suppressed Hpo-mediated phosphorylation of Yki (Supplementary Fig. 3d). These results suggest that Hpo/Wts and PRP4K phosphorylate Yki in distinct subcellular compartments with Hpo/Wts acting mainly in the cytoplasm and PRP4K in the nucleus.

**PRP4K phosphorylates Yki at S111 and S250.** Wts restricts Yki nuclear localization though phosphorylating Yki on multiple sites including S111, S168, and S250[19,20,22,32]. Phosphorylation of Yki S168 promotes its association with 14-3-3 and cytoplasmic sequestration, whereas phosphorylation of S111 and S250 restricts Yki nuclear localization independent of 14-3-3[22]. Interestingly,

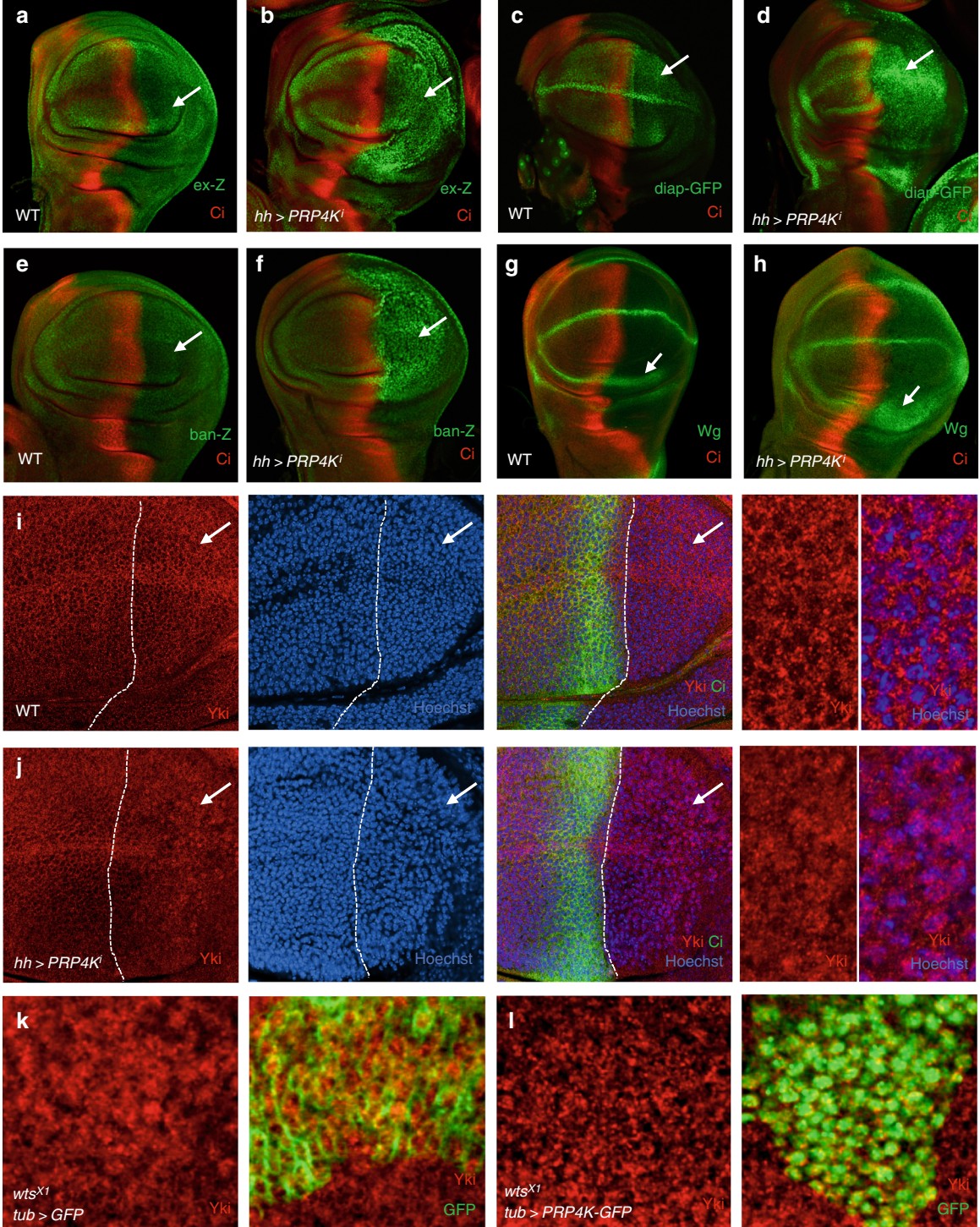

**Fig. 2** PRP4K restricts Yki nuclear localization and inhibits Hpo target gene expression. **a–h** Late third instar wing imaginal discs of control (**a**, **c**, **e**, **g**) or expressing *UAS-PRP4K-RNAi* with *hh-Gal4* (**b**, **d**, **f**, **h**) were immunostained to show the expression of Ci (red) and *ex-lacZ* (green in **a**, **b**), *diap-GFP3.5* (green in **c**, **d**), *ban-lacZ* (green in **e**, **f**), or Wg (green in **g**, **h**). Arrows indicate posterior compartments marked by the lack of Ci expression. Of note, *hh-gal4* is expressed in posterior compartment cells. **i–j** Late third instar wing imaginal discs of control (**i**) or expressing *UAS-PRP4K-RNAi* with *hh-Gal4* (**j**) were immunostained to show the expression Yki (red), Ci (green), and a nuclear dye Hoechst (blue). Arrows indicate the enlarged regions shown in the right most panels. **k–l** Late third instar wing imaginal discs bearing *wts* mutant clones that express either *UAS-GFP* (**k**) or *UAS-PRP4K-GFP* (**l**) under the control of *tub-Gal4* were immunostained for Yki and GFP. *wts* mutant clones are marked by GFP expression (green in the right panels). Of note, GFP and PRP4-GFP mark the cytoplasm and nucleus, respectively

mutating S168 to Ala (YkiS168A) did not affect PRP4K-induced mobility shift of Yki, whereas mutating S111 and S250 (Yki2SA) or mutating all three sites (Yki3SA) abolished PRP4K-induced mobility shift in S2 cells (Supplementary Fig. 3e–g), as well as in eye discs (Supplementary Fig. 3h–k). PRP4K RNAi in eye discs resulted in reduced phosphorylation of Yki and YkiS168 but had no effect on Yki2SA or Yki3SA (Supplementary Fig. 3h–k). Both gain- and loss-of-PRP4K function did not affect the signal detected by a phospho-specific antibody (pS168) that recognized Yki phosphorylated on S168 (Fig. 3d; Supplementary Fig. 3l–m)[19]. By contrast, Wts RNAi diminished Yki phosphorylation at S168 in eye discs (Supplementary Fig. 3m). In addition, Wts RNAi

resulted in a more dramatic reduction of Yki phosphorylation compared with PRP4K RNAi, and Yki phosphorylation was further reduced by combined knockdown of Wts and PRP4K (Supplementary Fig. 3n). These results suggest that PRP4K phosphorylates Yki on S111 and S250, but not on S168, and that Wts contributes more toward phosphorylating Yki than PRP4K.

To gain further evidence that PRP4K phosphorylates Yki on S111 and S250, HA-tagged wild type and kinase dead PRP4K (HA-PRP4K and HA-PRP4K[KR]) were immunopurified from S2 cells and incubated with purified GST fusion proteins containing Yki fragments that harbor either wild type or mutated S111, S168, or S250 (Fig. 3e) in the presence of [γ-$^{32}$p]-ATP. As shown in

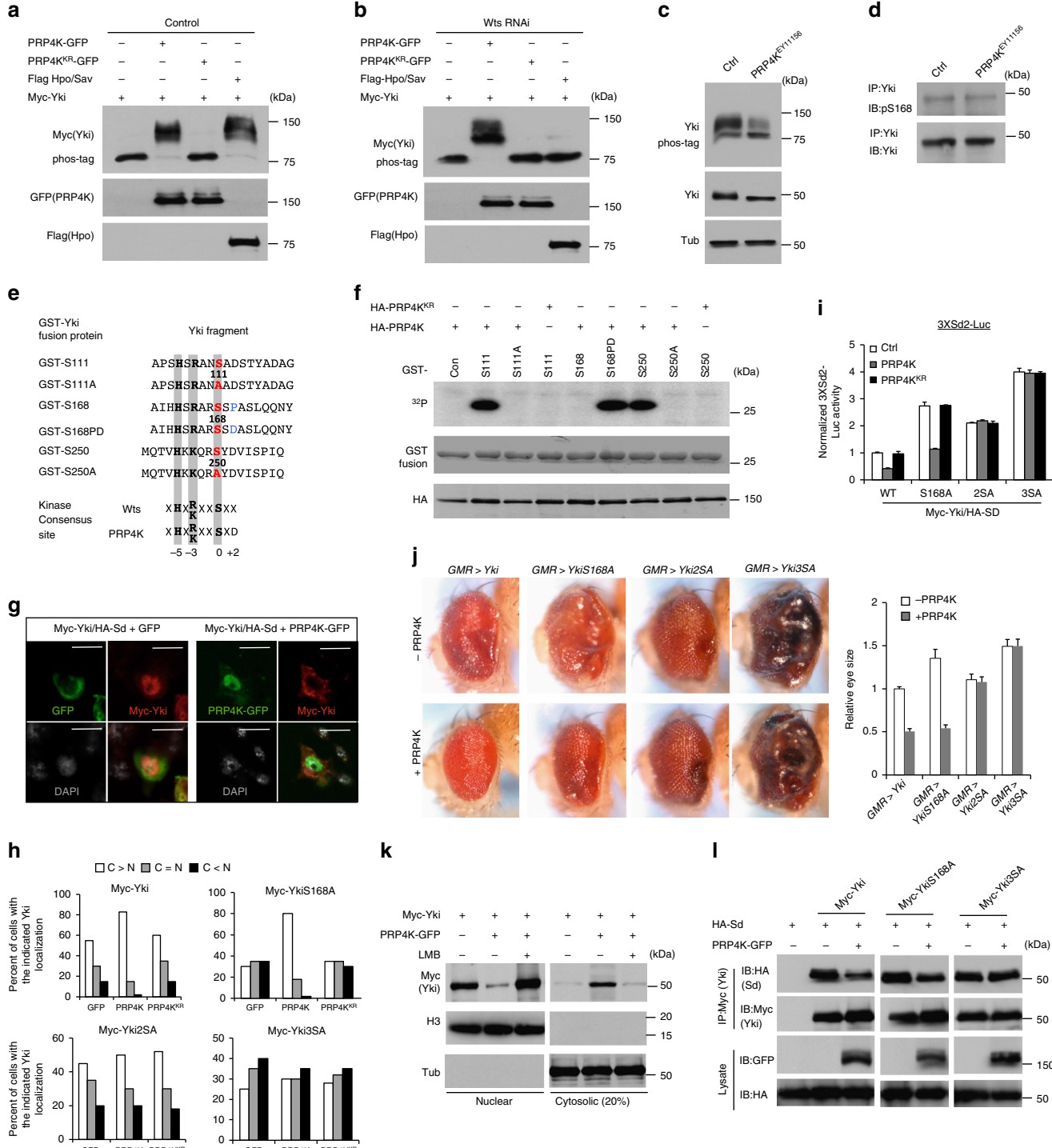

Fig. 3f, HA-PRP4K phosphorylated S111 and S250 but not S168, whereas HA-PRP4K[KR] did not phosphorylate any of these sites in the GST fusion proteins. All the three sites conform the Wts/Lats1/2 phosphorylation consensus: HXK/RXS/TX where X represents any amino acid (Fig. 3e)[14]. However, while S111 and S250 contain an acid residue (Asp) at the +2 position, S168 contains Pro at this position, which confers binding to 14-3-3 after S168 is phosphorylated by Wts. Interestingly, conversion of Pro to Asp (GST-S168PD) allowed this site to be phosphorylated by PRP4K (Fig. 3e, f). Hence, PRP4K phosphorylates a subset of Wts sites in Yki, and PRP4K and Wts/Lats1/2 have different selectivity toward their phosphorylation sites; although PRP4K prefers an acidic residue at the +2 position, Wts/Lats1/2 can accommodate Pro at this position.

In S2 cells, PRP4K but not PRP4K[KR] inhibited nuclear localization of Myc-tagged wild type Yki (Myc-Yki) and its S168A mutant (Myc-YkiS168A); however, PRP4K did not affect the subcellular localization of Myc-Yki2SA and Myc-Yki3SA (Fig. 3g, h). Furthermore, PRP4K suppressed a Sd-dependent luciferase (3XSd2-luc) reporter activity stimulated by Yki and YkiS168A but failed to inhibit Yki2SA- or Yki3SA-induced 3XSd2-luc expression (Fig. 3i). Overexpression of PRP4K suppressed eye overgrowth induced by excessive Yki or YkiS168A but not by Yki2SA or Yki3SA (Fig. 3j). Hence, PRP4K inhibits Yki nuclear localization and activity by phosphorylating S111/S250.

Our previous study revealed that phosphorylation of Yki at S111 and S250 restricted Yki nuclear localization through nuclear export[22]. Consistent with this, LMB treatment blocked the ability of PRP4K to restrict Yki nuclear localization (Fig. 3k). PRP4K inhibited the binding of Yki to Sd depending on S111/S250 phosphorylation (Fig. 3l), which may contribute to its ability to restrict Yki nuclear localization because binding to Sd facilitates Yki nuclear localization[21]. In addition, inhibition of Yki/Sd interaction may also contribute to the ability of PRP4K to inhibit Hpo pathway target gene expression.

**PRP4K regulates mammalian Hpo signaling independent of Lats1/2.** We next determined whether PRP4K played a conserved role in the regulation of Hpo pathway. Transfection of human PRP4K but not its kinase dead form, PRP4K[KR] (K718R), into HEK293 cells induced phosphorylation of co-transfected Yap (Fig. 4a). Deletion of both Lats1 and Lats 2 (Lats1/2 −/−)[17] abolished Mst1-induced but not PRP4K-induced Yap phosphorylation (Fig. 4b), suggesting that PRP4K-mediated phosphorylation of Yap is independent of Lats1/2.

Overexpression of PRP4K but not PRP4K[KR] resulted in nuclear exclusion of Yap in both wild type and Lats1/2 −/− HEK293 cells (Fig. 4c; Supplementary Fig. 4a-c). Starvation induced Yap phosphorylation and nuclear-to-cytoplasmic translocation of Yap[33], both of which were suppressed by siRNA-mediated depletion of PRP4K (Fig. 4d, e). Furthermore, transfection of PRP4K but not it kinase dead form restored Yap phosphorylation in response to starvation (Fig. 4d). Hence both loss- and gain-of-function studies demonstrate that PRP4K plays a conserved role in the regulation of Yap phosphorylation and subcellular localization, and it does so independent of Lats1/2.

A previous study showed that starvation-induced Yap phosphorylation and nuclear-to-cytoplasmic translocation depended on Lats1/2[17]. Consistent with this, we found that starvation-induced Yap phosphorylation was diminished in Lats1/2−/− HEK293 cells similar to HEK293 cells in which PRP4K was depleted by siRNA (Fig. 4f). Furthermore, depletion of PRP4K in Lats1/2 −/− HEK293 cells completely blocked starvation-induced Yap phosphorylation (Fig. 4f). These results suggest that efficient phosphorylation of Yap in response to starvation requires both Lats1/2 and PRP4K.

Lats1/2 phosphorylates Yap on at least five Ser residues[14], among which the S127 site is equivalent to the S168 of Yki and has Pro at the +2 position whereas the other four sites (S61, S109, S164, and S384) are similar to the S111/S250 site of Yki in that they have acid residues (D/E) or T at this position (Fig. 4g). Similar to the S168A mutation that did not affect Yki phosphorylation by PRP4K, the S127A mutation did not affect Yap phosphorylation by PRP4K (Fig. 4h, i). By contrast, mutating S61, S109, S164, and S384 (Yap[4SA]) or all five sites (Yap[5SA]) abolished PRP4K-mediated phosphorylation of Yap (Fig. 4j, k). Luciferase reporter assay indicated that activity of Yap and Yap[S127A] but not of Yap[4SA] or Yap[5SA] were inhibited by PRP4K (Fig. 4l). Further, PRK4K inhibited the binding of TEAD4 to Yap but not to Yap[5SA] (Fig. 4m,n). Taken together, these results suggest that PRP4K regulates mammalian Hpo signaling pathway by phosphorylating a subset of Lats1/2 sites on Yap.

Another mammalian homolog of Yki is Taz[34]. We found that PRP4K phosphorylated and inhibited the nuclear localization and activity of co-transfected Taz (Supplementary Fig. 5a, c, d). Mutating the four Lats sites to Ala (Taz[4SA])[35] abolished PRP4K-mediated phosphorylation of Taz (Supplementary Fig. 5b). In addition, the nuclear localization and activity of Taz[4SA] were no longer inhibited by PRP4K (Supplementary Fig. 5c, d), suggesting that PRP4K can phosphorylate and inhibit Taz.

---

**Fig. 3** PRP4K inhibits Yki nuclear localization and activity by phosphorylating Yki on Ser111/250. **a**, **b** S2 cells treated with control or Wts dsRNA were transfected with the indicated constructs. Cell extracts were separated on phos-tag conjugated (as indicated) or regular SDS-PAGE and immunoblotted with the indicated antibodies. **c**, **d** Cell extracts from eye discs carrying control (ctrl) or PRP4K mutant clones were separated on phos-tag conjugated or regular SDS-PAGE and immunoblotted with the indicated antibodies. **e** Sequence alignment of wild type and mutated Wts phosphorylation sites included in the indicated GST-Yki fusion constructs. The phospho acceptor sites are highlighted in red. "X" in the consensus sequence denotes any amino acid. **f** Autoradiograph (top panel) of an in vitro kinase assay using GST-Yki fusion proteins containing the indicated Wts phosphorylation sites and immunopurified HA-PRP4K or HA-PRP4K[KR] in the presence of [γ-$^{32}$p] ATP. Coomassie blue staining (middle) and western blot (bottom) show that equal amounts of GST fusion proteins and HA-PRP4K were used. **g** Immunostaining of S2 cells transfected with the indicated constructs. **h** Quantification of subcellular localization of Myc-tagged wild type and mutant Yki transfected into S2 cells with GFP, PRP4K-GFP, or PRP4K[KR]-GFP. C>N: higher Myc signal intensity in cytoplasm than in nucleus; C = N: equal Myc signal intensity in cytoplasm and nucleus; C < N: higher Myc signal intensity in nucleus than in cytoplasm. n = 100 cells were examined for each experimental condition. **i** 3XSd2-luc reporter assay in S2 cells transfected with the indicated constructs. Data are means ± s.d. from three independent experiments. **j** Drosophila adult eyes expressing the indicated Yki transgenes in the absence or presence of UAS-PRP4K. Quantification of eye size for the indicated genotypes. Data are means ± s.d. from three independent experiments. N ≥ 5 for each genotype. **k** S2 cells transfected with the indicated constructs were treated with or without LMB. Cell extracts were fractionated into nuclear and cytosolic fractions, followed by western blot analysis with the indicated antibodies. H3: histone 3. Tub: tubulin. **l** S2 cells were transfected with the indicated constructs. Cell lysates were immunoprecipitated with a Myc antibody, followed by western blot analysis with HA and Myc antibodies, or directly subjected to western blot analysis with GFP and HA antibodies. Scale bars, 10 μm (**g**)

**PRP4K inhibits proliferation and invasiveness of breast cancer cells.** Yap/Taz has been implicated as oncogene in a wide variety of human cancers[5]. Indeed, high Yap expression levels correlated with poor survival in triple-negative breast cancer patients (Fig. 5a). Consistence with this, inactivation of Yap/Taz affected the growth and invasion of triple-negative breast cancer (TNBC) cells such as MDA-MB-231 cells[36–38]. Interestingly, high PRP4K

expression levels correlated with good prognosis in triple-negative breast cancer patients (Fig. 5b), consistent with its role in opposing Yap/Taz activity. Knockdown of PRP4K in MDA-MB-231 cells by two independent siRNAs decreased Yap phosphorylation (Fig. 5c), increased Yap protein levels (Fig. 5d) and the expression of Yap target genes including *CTGF*, *CRY61*, *ANKRD1*, and *AJUBA* (Fig. 5e)[37]. The increase of Yap target gene

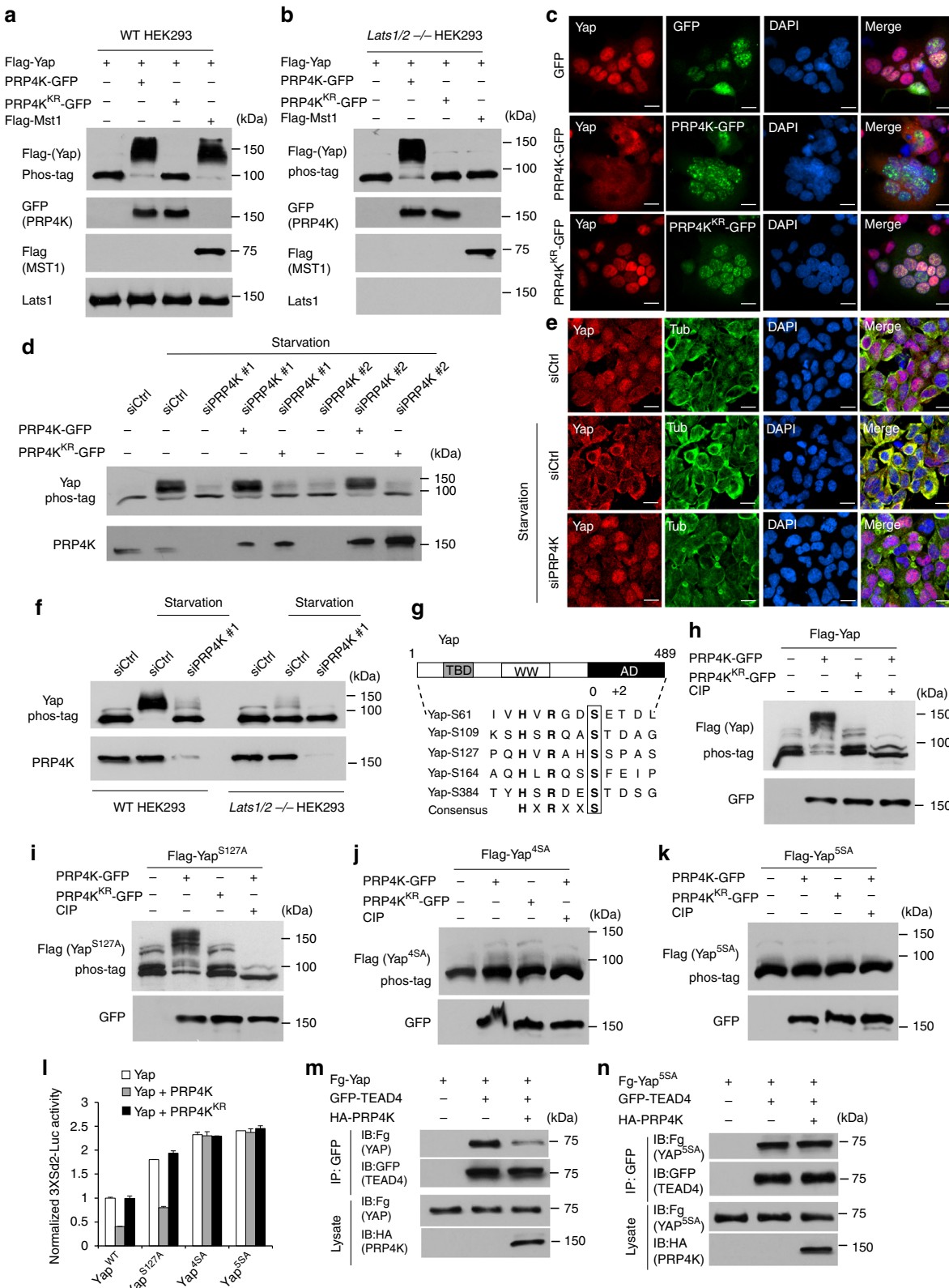

expression caused by PRP4K RNAi was reversed by Yap knock-down (Fig. 5f). Furthermore, PRP4K knockdown increased the proliferation, migration, and invasiveness of MDA-MB-231 cells (Fig. 5g–j). On the other hand, lentiviral infection of PRP4K but not PRP4K$^{KR}$ into MDA-MB-231 cells increased Yap phosphorylation (Fig. 5k), reduced Yap target gene expression (Fig. 5l), and decreased the proliferation and migration of the cancer cells (Fig. 5m–o). These results suggest that PRP4K can inhibit TNBC growth through regulating the Hpo signaling pathway.

## Discussion

Although it has been well established that phosphorylation of Yki/Yap/Taz by a cytoplasmic kinase cassette involving Hpo/MST1/2 and Wts/Lats1/2 impedes its nuclear translocation and activity, how Yki/Yap/Taz is regulated in the nucleus is still largely unexplored. Here we identified a nuclear kinase PRP4K that plays an evolutionarily conserved role in modulating the Hpo pathway activity. We demonstrated that PRP4K acts in parallel with the Hpo/MST1/2-Wts/Lats1/2 kinase cascade and phosphorylates Yki/Yap/Taz in the nucleus on a subset of Wts/Lats1/2 sites (S111 and S250 on Yki), leading to its nuclear exclusion in a manner depending on nuclear export.

Previous studies suggest that Wts/Lats1/2 phosphorylates Yki/Yap/Taz in the cytoplasmic compartment[39,40], which promotes its association with 14-3-3 and cytoplasmic sequestration. Yki/Yap/Taz phosphorylation is likely to be dynamic due to the action of Yki/Yap/Taz phosphatase[41,42]. As a consequence, dephosphorylated Yki/Yap/Taz may enter the nucleus to cause leaky activation of Hpo pathway target genes. However, we found that Yki/Yap/Taz was phosphorylated by PRP4K in the nucleus, which promoted its nuclear export (Fig. 6). Hence, the nuclear kinase activity of PRP4K provides a second line of defense to prevent aberrant nuclear accumulation and activation of Yki/Yap/Taz. This fail-safe mechanism is likely to play an important role to restrict cell growth under conditions where Yki/Yap/Taz levels are elevated, as seen in GMR>Yki eye imaginal discs and in MDA-MB-231 cells where NF2, an upstream regulator of the Hpo pathway, was deleted[37]. However, under physiological condition where the core Hpo kinase cassette is intact, Wts/Lats1/2 appears to play a more dominant role in phosphorylating and inhibiting Yki/Yap/Taz, which explains why loss of Wts but not PRP4K resulted in dramatic tissue overgrowth.

Elevation of Yap/Taz expression has been observed in many types of human cancer[5], raising the possibility that PRP4K may play a role in the progression of these cancers. The observation that high levels of PRP4K expression correlate with good prognosis of triple-negative breast cancer patients supports a role of PRP4K in tumor suppression. Further study is needed to fully explore the function of PRP4K in cancer progression and how PRP4K is regulated.

## Methods

***Drosophila* genetics and transgenes.** Mutant stocks: EY11156 (BL# 20662) is a P-element insertional allele of PRP4K/CG7028 (Flybase). It is a lethal mutation caused by an insertion of P{EPgy2} at the 5' region of the CG7028 coding sequence. P{EPgy2} was inserted in the opposite orientation relative to CG7028 transcription. wts$^{X1}$ is strong allele of wts[12]. Transgenic RNAi lines: UAS-PRP4K-RNAi-1 (VDRC#107042), UAS-PRP4K-RNAi-2 (VDRC#27698), UAS-PRP4K-RNAi-3 (BL#55640), and UAS-Wts-RNAi (VDRC#106174). Transgenes: UAS-Yki, UAS-YkiS168A, UAS-Sd-GA, and diap-GFP3.5[21], UAS-Yki2SA and UAS-Yki3SA[22], UAS-InR$^{AC}$ (BL#8263), bam-lacZ[43], and ex-lacZ[44]. Gal4 drivers: GMR-Gal4[45] and hh-Gal4[21]. Drosophila eyes containing control or PRP4K$^{EY11156}$ mutant clones with or without Yki overexpression were generated using the following genotypes:

 y w ey-Flp; FRT80A/RpS14[4] w$^+$ FRT80A, y w ey-Flp; EY11156 FRT80A/RpS14 [4] w$^+$ FRT80A, y w ey-Flp; GMR-Gal4 UAS-Yki; FRT80A/RpS14[4] w$^+$ FRT80A, y w ey-Flp; GMR-Gal4 UAS-Yki; EY11156 FRT80A/RpS14[4] w$^+$ FRT80A.

 wts clones expressing UAS-GFP or UAS-PRP4K-GFP: y w UAS-GFP hs-Flp; tubulin-GaL4/+; FRT82B tubulin-Gal80/FRT82B wts$^{X1}$ y w hs-Flp; tubulin-GaL4/ UAS-PPR4K-GFP; FRT82B tubulin-Gal80/FRT82B wts$^{X1}$.

**DNA constructs.** pUAST constructs: UAS-Myc-Yki, UAS-Myc-YkiS168A, UAS-Myc-Yki2SA, UAS-Myc-Yki3SA, UAS-Flag-Hpo, UAS-Sav, and UAS-HA-Sd were previously described[10,21,22]. Myr-Yki-GFP contained a myristoylation signal MGNKCCSKRQ and GFP tag at the N- and C-terminus, respectively. Drosophila PRP4K cDNA clone was obtained from DGRC (Flybase). PRP4K$^{KR}$ (K617R) variant was generated by PCR-mediated site-directed mutagenesis. GFP or HA tag was inserted at the C-terminus of PRP4K or PRP4K$^{KR}$. The constructs were subcloned into the pUAST vector digested with NotI and XbaI. pcDNA3.1 constructs expressing Flag-Yap, Flag-YAPS127A, Flag-YAP5SA, HA-Taz, and HA-Taz4SA are a gift from Dr. Yingzi Yang (Harvard School of Dental Medicine). Flag-YAP4SA was derived from Flag-YAP5SA by PCR-mediated site-directed mutagenesis to convert A127 back to S127. Human PRP4K was obtained from Dharmacon. PRP4K$^{KR}$ (K718R) was generated by PCR-mediated site direct mutagenesis. GFP tag was inserted at the C-terminus of PRP4K or PRP4K$^{KR}$ and the constructs were subcloned into pcDNA3.1 for transient transfection experiments or FUXW for lentiviral infection experiments. To make GST-Yki fusion constructs, synthetic DNA oligos corresponding to the coding sequences for Yki aa105-122, aa160-181, aa 237-260, and their SA or PD mutants (S111A, S168PD, and S250A) were annealed and subcloned into pGEX 4T-1 vector digested with EcoRI and XhoI.

**Cell culture, transfection, and lentiviral production.** S2 cells were cultured in Schneider's Drosophila Medium (Life Technologies) with 10% fetal bovine serum (GE Healthcare), penicillin (100 U/ml; Life Technologies), and streptomycin (100 mg/ml; Life Technologies) at 24 °C. Transfection of S2 cells was performed using Calcium Phosphate Transfection Kit (Specialty Media) following manufacturer's instruction. A ubiquitin-Gal4 construct was co-transfected with pUAST constructs to drive UAS transgene expression in S2 cells. HEK293A and MDA-MB-231 cells were obtained from the American Type Culture Collection (ATCC). Cell line authentication utilizing Short Tandem Repeat (STR) profiling were performed with PowerPlex® 21 System (Promega, USA), which allowed for detection of 21 loci. The PCR amplification of DNA templates were analyzed by capillary electrophoresis with genetic analyzer 3130 (ABI, USA) and allele call was performed by Gene-Mapper software V3.2.1. The markers were used to compare with the accessible database archived by ATCC. The cells were cultured in DMEM (Sigma-Aldrich)

**Fig. 4** PRP4K plays a conserved role in mammalian Hpo signaling pathway. **a, b** Wild type or Lats1/2 −/− HEK293A cells were transfected with the indicated constructs. Cell extracts were separated on phos-tag conjugated or regular SDS-PAGE and immunoblotted with the indicated antibodies. **c** HEK293A transfected with the indicated constructs were immunostained with YAP and GFP antibodies and DAPI. **d** HEK293A cells were transfected with control or PRP4K siRNA in the absence or presence of PRP4K/PRP4K$^{KR}$-GFP and starved for 6 h prior to lysis. Cell extracts were separated on phos-tag conjugated or regular SDS-PAGE and immunoblotted with the indicated antibodies. **e** HEK293A cells were transfected with control or PRP4K siRNA and starved for 6 h before fixation and immunostaining with Yap and tubulin antibodies. **f** Wild type and Lats1/2−/− HEK293 cells were transfected with control or PRP4K siRNA and starved for 6 h prior to lysis. Cell extracts were separated on phos-tag conjugated or regular SDS-PAGE and immunoblotted with the indicated antibodies. **g** Schematic drawing of Yap with Lats1/2 phosphorylation sites aligned underneath. TBD: TEAD binding domain. WW: WW domain. AD: activation domain. **h–k** Cell extracts from HEK293 cells transfected with the indicated constructs were separated on phos-tag conjugated or regular SDS-PAGE and immunoblotted with the indicated antibodies. **l** 3XSd2-luc reporter assay in HEK293 cells transfected with the indicated constructs. Data are means ± s.d. from three independent experiments. **m, n** HEK293 cells were transfected with the indicated constructs. Cell lysates were immunoprecipitated with a GFP antibody, followed by western blot analysis Flag and GFP antibodies, or directly subjected to western blot analysis with Flag and HA antibodies. Scale bars, 10 µm (**c, e**)

containing 10% BCS (ATCC), and transfected using GenJet Plus in vitro DNA transfection kit (SignaGen). For starvation experiments, HEK293A cells were cultured in DMEM (Sigma-Aldrich) without serum for 6 h two days after transfection. For lentivirus production, HEK293T-17 cells were seeded and transfected with *psPAX2*, *VSVG*, and *FUXW* expressing GFP, PRP4K-GFP, or PRP4K$^{K718R}$-GFP. The recombinant viruses were infected into MDA-MB-231 cells using the standard method.

**RNA interference**. For RNAi experiments in S2 cells, dsRNAs were generated using the MEGAscript High Yield Transcription Kit (Ambion). The following primers were used for generating the dsRNA targeting Wts:

TAATACGACTCACTATAGGGCACCCAGTTATTGTCG (F) and TAAACGAC TCACTATAGGGGTTCTTCATGGAGCAGCACATAG (R). dsRNA targeting the coding sequence of Luciferase was used as a control. For RNAi in mammalian cells, siRNA was transfected into cells using Lipofectamine RNAi-MAX (Life Technologies) in antibiotics-free medium according to manufacturer's instruction. Oligonucleotides of siRNA duplexes were purchased from Dharmacon as follow: siPRP4K#1 (J-004074-11), siPRP4K#2 (J-004074-12), and siYap1 (J-012200-07 and J-012200-08), and siControl (D-001810-03-05).

**Immunoprecipitation, western blot, and immunostaining**. For immunoprecipitation assay, cells were harvested and washed twice with PBS after transfection for

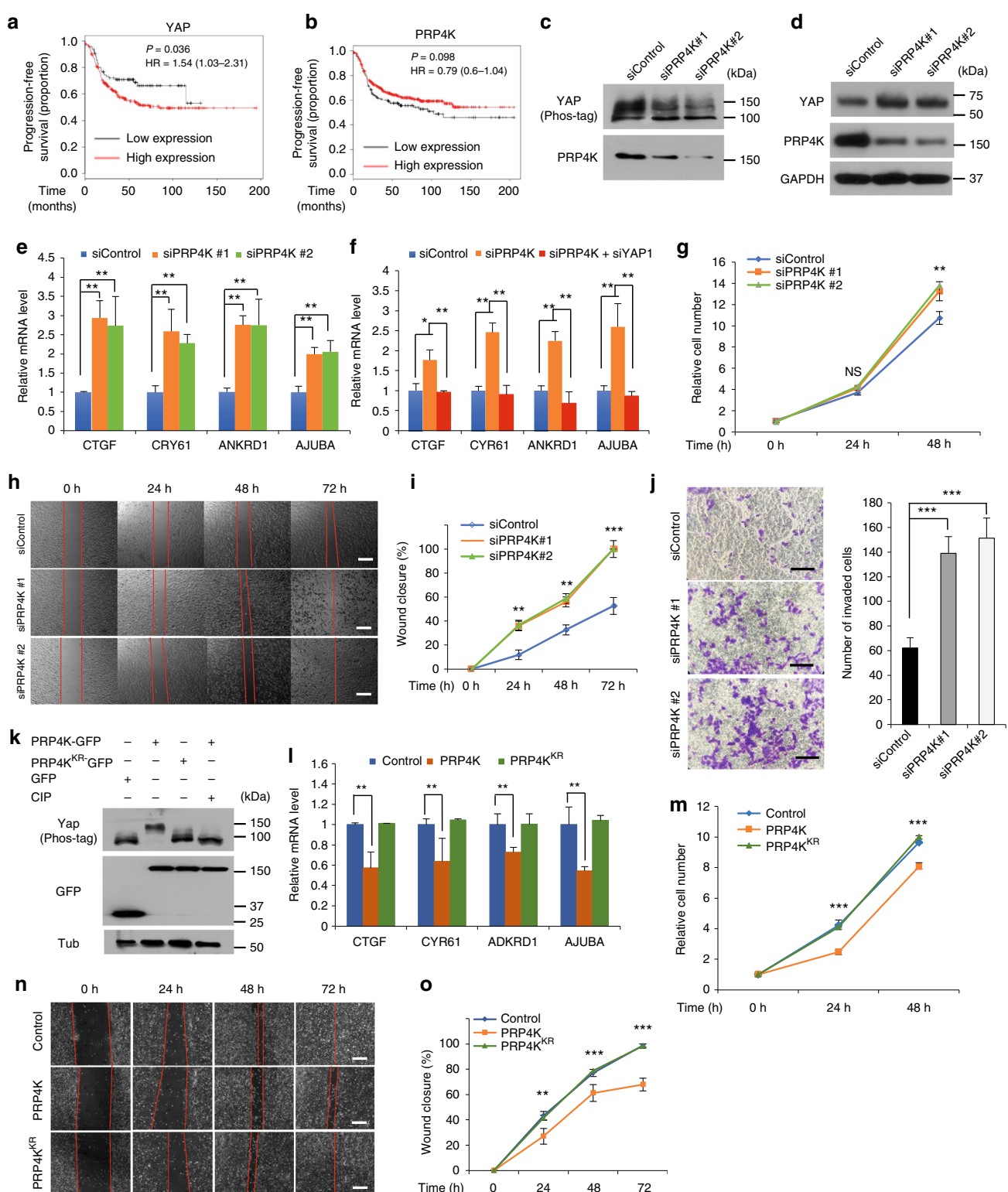

**Fig. 5** PRP4K regulates growth and invasiveness of breast cancer cells. **a**–**b** Kaplan–Meier graphs of progression-free survival analysis showing that high YAP expression correlated with poor prognosis (**a**) whereas high PRP4K (PRPF4B) expression correlated with good prognosis (**b**) in triple-negative breast cancer patients. Data were obtained from KMPLOTdatabase (http://kmplot.com/). **c, d** PRP4K depletion decreased Yap phosphorylation (**c**) and increased YAP protein level (**d**) in MDA-MB-231 cells. **e, f** PRP4K depletion in MDAMB231 cells increased Hpo pathway target gene expression (**e**), which was reversed by Yap RNAi (**f**). mRNAs of the indicated Hpo target genes were measured by RT-qPCR. Data are means ± s.d. from three independent experiments. * P < 0.05, ** P < 0.01 (Student's t-test). **g** PRP4K depletion increased MDA-MB-231 cell proliferation. Cell numbers were measured by the WST-1 assay. Data are means ± s.d. from three independent experiments. ** P < 0.01 (Student's t-test). NS, not significant. **h** Wound healing assay of MDA-MB-231 transfected with the indicated siRNA. **i** Quantification of wound closure at the indicated time points. Data are means ± s.d. from three independent experiments. ** P < 0.01, *** P < 0.001 (Student's t-test). **j** Transwell invasion assay of MDAMB-231 transfected with the indicated siRNA. Data are means ± s.d. from three independent experiments. ** P < 0.01, *** P < 0.001 (Student's t-test). **k** Western blot analysis of MDA-MB-231 cells infected with the indicated recombinant lentiviruses. **l** mRNAs of Hpo target genes from MDA-MB-231 cells infected with the indicated lentiviruses were measured by RT-qPCR. Data are means ± s.d. from three independent experiments. ** P < 0.01 (Student's t-test). **m** Relative cell numbers of MDA-MB-231 cells infected with the indicated lentiviruses were measured by the WST-1 assay at different time points after infection. Data are means ± s.d. from three independent experiments. *** P < 0.001 (Student's t-test). **n** Wound healing assay of MDA-MB-231 infected with the indicated lentiviruses. **o** Quantification of wound closure of lentiviral infected MDA-MB-231at the indicated time points. Data are means ± s.d. from three independent experiments. ** P < 0.01, *** P < 0.001 (Student's t-test). Scale bars, 100 μm (**h**, **n**) or 20 μm (**j**)

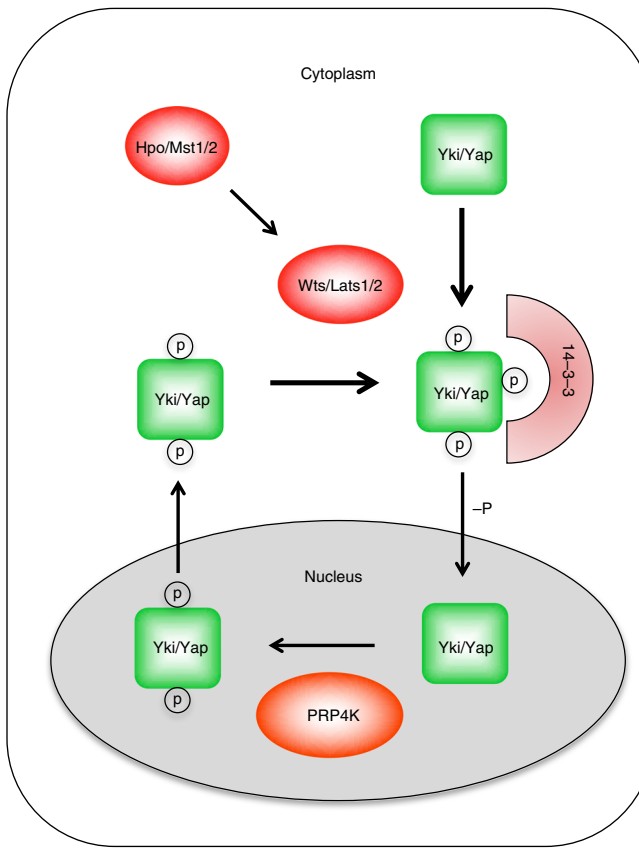

**Fig. 6** Model for how PRP4K regulates Hpo signaling. PRP4K phosphorylates Yki/Yap in the nucleus to promote its cytoplasmic translocation whereas the Hpo/Mst1/2-Wts/Lats1/2 kinase cassette phosphorylates Yki/Yap to promote its binding to 14-3-3, resulting in sequestration of Yki/Yap in the cytoplasm

48 h and then lysed on ice for 30 min with lysis buffer containing 1 M Tris pH8.0, 5 M NaCl, 1 M NaF, 0.1 M $Na_3VO_4$, 1% NP-40, 10% Glycerol, and 0.5 M EDTA (pH8.0). Cell lysates were incubated with protein A–Sepharose beads (Thermo scientific) for 1 h at 4 C° to eliminate non-specific binding proteins. After removal of the protein-A beads by centrifugation, the cleared lysates were incubated with Myc (HA or Flag) antibody for 2 h or overnight. The complexes were collected by incubation with protein A–Sepharose beads for 1 h at 4 C°, followed by centrifugation. The immunoprecipitates were then washed three times for 5 min each with lysis buffer and were separated on SDS-PAGE. West blot was carried out using standard protocol. Uncropped blots are shown in Supplementary Fig. 6. Ten percent acrylamide phospho-tag gel with 50 μM phospho-tag acrylamide were used to examine Yki, Yap, and Taz phosphorylation[31,46]. Immunostaining of imaginal

discs was carried out using standard protocal[47]. For immunostaining of cultured cells, cells were seeded on LAB-TEK chamber slides, transfected with indicated constructs and stained using standard protocols. Antibodies used: rabbit anti-sed: (ICN Biomedicals, 55976, 1:500), mouse anti-Wg (DSHB, 4D4-s, 1:50), rat anti-Ci (DSHB, 2A1-s, 1:50), rabbit anti-Yki (1:1000 for IB, 1:500 for IP)[16], Yki S168 phospho-specific antibody (pS168, 1:500)[19], mouse anti-GFP (Santa cruz, sc-9996, 1:1000), rabbit anti-GFP (Invitrogen, G10362, 1:500), mouse anti-Myc (Santa Cruz, sc-40, 1:1000), mouse anti-Flag (Sigma, F3165, 1:5000), mouse anti-HA (Santa Cruz sc-7392, 1:1000), rabbit anti-HA (Santa cruz, sc-805, 1;500), rabbit anti-Lats1 (Cell signaling, 9153, 1:500), mouse anti-Yap (Santa cruz, sc-101199, 1:1000), rabbit anti-Yap (Abcam, ab52771, 1:500), rabbit anti-PRP4K (Cell signaling, 8577, 1:500), mouse anti-Histon-H3 (Santa cruz, sc-56616, 1:1000), and mouse anti-tubulin (Sigma, T6199, 1:1000 for IB, 1:500 for immunostaining).

**RT-qPCR**. Total RNA was extracted using RNeasy Plus Mini Kit (Qiagen #74134), and cDNA was synthesized using the iScript cDNA synthesis kit (Bio-Rad). RT-qPCR was performed using iQ SYBR Green System (Bio-Rad) and a Bio-rad CFX96 real-time PCR system.
36B4 expression level was used a normalization control. Primer sequences used are:
CTGF, CATCTTCGGTGGTACGGTGT (F) and TTCCAGTCGGTAAGCCGC (R); CYR61, CGGGTTTCTTTCACAAGGCG (F) and TGAAGCGGGCTC CCTGTTTTT (R); ANKRD1, GCCATGCCTTCAAAATGCCA (F) and AGAACTGTGCTGGGAAGACG (R) AJUBA, TACCAGGACGAGCTAACAGC (F) and TACAGGTGCCGAAGTAGTCC (R) 36B4, GGCGACCTGGAAGT CCAACT (F) and CCATCAGCACCACAGCCTTC (R).

**In vitro kinase assay**. GST-fusion proteins were expressed in bacteria, purified with glutathione bead, and mixed with 0.1 mM ATP, 10 mCi $\gamma$-$^{32}$p-ATP, and immune-purified HA-PRP4K or HA-PRP4KKR expressed in S2 cells. After incubation at 30 °C for 1.5 h in the reaction buffer, the reactions were stopped by adding 4 × SDS loading buffer and boiled at 100 °C for 5 min, the phosphorylation of GST-fusion proteins were analyzed by autoradiography after SDS-PAGE.

**Luciferase reporter assay**. To measure the activity of Yki/YAP/TAZ, cells were seeded in 24-well plates and co-transfected with a Sd (Scalloped)-dependent luciferase reporter construct (3XSd2-luc) with pTK-Renilla and effector plasmids. The luciferase activities were analyzed using a dual-luciferase reporter assay kit (Promega) according to manufacturer's instruction.

**Cell proliferation assay**. MDAMB231 cells were seeded and transfected with 50 μM PRP4K siRNA or control siRNA. Twenty-four hours after transfection, cells were seeded into 96-well plates. Cell numbers were determined using WST-1 cell proliferation reagent (Cat: 5015944001, Sigma-Aldrich) at indicated time points.

**Wound healing assay**. MDAMB231 cells were seeded and transfected with 50 μM PRP4K siRNA or control siRNA. Twenty-four hours after transfection, the cells were seeded into 6-well plates with 1% FBS with 100% confluence. One yellow pipette tip was used to make a straight scratch. The wound distance was measured at indicated time points and normalized with starting time point. Percentage wound recovery was expressed as: [1-(Width of the wound at a given time/width of the wound at $t = 0$)] × 100%

**Transwell invasion assay**. The transwell invasion assay was performed with two-chamber plates (#3422, Corning). For the transwell assay, $1 \times 10^5$ MDAMB231

control cells and PRP4K knocking-down cells were seeded into the top chamber. To stimulate the invasion, complete medium was added to the bottom wells. After 24 h incubation at 37 °C for invasion assay, cells in the upper chamber were carefully removed and the cells that had passed through the membrane were fixed and stained with Crystal Violet Staining Solution. Cellular quantification was analyzed in three fields with ×100 magnification under microscope. The quantification was based on cell number counting in each field.

**Progression-free survival (PFS) data analysis**. The PFS survival data of PRP4K and YAP was generated from KMPLOT online analysis database (http://kmplot.com/analysis/index.php?p=service&cancer=breast). The gene affy IDs are 224894_at and 202127_at for YAP and PRP4K, respectively. 'Auto selected best cutoff' and 'basal type' were selected for triple-negative breast cancer analysis.

**Statistics and reproducibility**. All experiments were performed for at least two independent times unless noted otherwise. Statistical analysis between groups was performed by two-tailed Student's $t$-test to determine significance when only 2 groups were compared. $P$-values of less than 0.05 and 0.01 were considered significant. Error bars on all graphs are presented as the s.d.

**Data Availability**. All relevant data are available from the authors.

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

## Acknowledgements

We thank Drs. Yingzi Yang and Kunliang Guan for providing reagents, Bloomington and VDRC stock centres for fly stocks, and DSHB for antibodies. This work was supported by

grants from NIH (GM118063) and Welch Foundation (I-1603) to Jin Jiang (J.J.) and the Joint Fund of the National Natural Science Foundation of China (U1604190) to Jian Zhu. J.J. is a Eugene McDermott Endowed Scholar in Biomedical Science at UTSW.

## Author Contributions

Y.C., J.Z., Y.H., S.L., and B.W. performed the experiments. Y.C., J.Z., Y.H., S.L., and J.J. analyzed the data, Y.C., J.Z., Y.H., and J.J. designed the experiments. J.J. wrote the manuscript.

## Additional information

**Competing interests:** The authors declare no competing interests.

