## [Peer Review File · Nature Communications]

Reviewer #1 (Remarks to the Author):

In this study, Cho et al discovered that PRP4K, a protein kinase that has been implicated in splicing, regulates Yki/Yap activity. In *Drosophila*, knockdown of PRP4K enhances the Yki overexpression phenotypes, both tissue growth and target gene expression, while overexpression of PRP4K suppresses the Yki overexpression phenotypes. PRP4K is found to act downstream or parallel to Hpo. Wts inhibits Yki by direct phosphorylation, including phosphorylation of S168 that creates 14-3-3 binding and induces Yki cytoplasmic localization. Interestingly, PRP4K also directly phosphorylates Yki on the Wts consensus sites S111 and S259, but not on the primary Wts site S168. Through an unknown mechanism, PRP4K phosphorylation increases Yki cytoplasmic localization, therefore inhibiting Yki activity. Evidence is presented to show that Wts and PRP4K phosphorylate Yki in the cytoplasm and nucleus, respectively. Further studies in mammalian cells suggest a similarly conserved function and mechanism of PRP4K in YAP and TAZ regulation.

This study provides interesting data to support a model that PRP4K can directly phosphorylate Yki/Yap on the Lats consensus sites, yet the PRP4K phosphorylation sites are not entirely overlapping with Wts/LATS. Besides the conventional Hippo pathway components, many genes have been implicated in Yap regulation in a "Hippo independent" manner, yet some of the reported new "regulators" do not stand further scrutiny. Available evidence overwhelmingly supports a model that Wts/LATS are probably the most important regulators of Yki/Yap. To appreciate a functional significance of PRP4K in Yki regulation, more evidence is needed to compare the relative contribution of PRP4K and Wts in Yki inhibition.

The phenotypes of PRP4K mutation and Wts mutation in tissue growth, Yki cellular localization and phosphorylation, and target gene expression should be compared side-by-side. This is very important, so the readers can appreciate the true physiological importance of PRP4K in Yki regulation.

Based on the proposed mechanism that Wts and PRP4K phosphorylate Yki on different sites in different subcellular compartment, double knockdown of Wts and PRP4K should be performed. The model predicts that double knockdown should produce additive or synergistic effects on Yki localization and tissue growth.

Fig.3f. The data is very clean. However, using very short peptides as kinase substrates could produce misleading information. Is it possible to repeat the experiments with full length Yki and show that S111 and S250 are the primary PRP4K sites?

Fig. S3h-i. The data is rather strange and inconsistent with data in Fig.3h. Fig.3h shows that YkiS168A is much more nuclear than Yki2SA that is almost identical to Yki wild type, suggesting S168 being the primary regulatory site in Yki. However, in Fig. S3h-I, patterns on phostag gel of YkiS168A is almost identical to wild type Yki while Yki2SA (Fig.S3j) abolishes the mobility shift, suggesting that S168 has little impact on Yki phosphorylation. How these data can be reconciled?

Fig.4d-e. The starvation-induced YAP mobility shift and localization should be compared with LATS1/LATS2 knockout cells to show the relative contribution of LATS and PRP4K in YAP regulation.

Reviewer #2 (Remarks to the Author):

This paper by Jiang and colleagues presents evidence for a new member of the Hippo pathway, the kinase PRP4K, that acts by phosphorylating Yki/YAP thereby promoting its nuclear export. The study is novel and significant as it relates to a new regulator of the important Hippo tissue growth control pathway. PRP4K has been previously found to play roles in RNA slicing and the spindle

assembly checkpoint, and more recently in EGFR membrane trafficking (Corkery et al., 2017) - so it appears to have pleiotropic functions - some discussion of these other roles and how the current role in the Hippo pathway might relate to these other findings would be helpful to the reader.

The study is well written and the data mostly clear, however it should be strengthened by quantification, particularly in instances where the effects seen are small.

Specific comments:

(1) It is now standard practise that effects observed on eye sizes should be quantified. This applies to Fig 1, Fig Supp 1 and Fig 3J.

(2) Figure 1D "PRP4K overexpression suppressed diap1-GFP3.5 expression and eye overgrowth induced by GMR>Yki" - This is not convincing - needs quantification of posterior vs anterior staining (where the GMR driver is not expressed) is needed for this and the other eye disc images.

(3) PRP4K- EY11156 mutant - This is an enhancer P line - where is it localised in the gene and how is it orientated? Some explanation is needed for this allele in the methods.

(4) Figure 1f - There seem to be different shades of red/orange - perhaps due to some heterozygous tissue remaining? To illustrate the growth advantage more clearly could also examine this in the larval eye disc to see if there is a clonal advantage of PRP4K at this stage, as is expected for Hippo pathway mutations.

(5) Supp Fig 2e-i - "Coexpression of PRP4K but not PRP4KKR, or knockdown of Yki suppressed the elevated ex-lacZ expression caused by hh>PRP4Ki" - Is this statement correct? Yki knockdown does look like it suppresses ex-lacZ upregulation? To make this analysis more convincing could quantify the staining in the posterior versus anterior wing domains. It should be stated in the sup figleg what the Ci staining detects.

(6) Supp Fig 3d - Seems like there is less overall Yki in the LMB Hpo lane - ie would expect that if Yki cant exit the nucleus Hpo-Wts phosphorylation of it would decrease, but there should be more unphosphorylated Yki (lower band)? Is there really less Yki here? If so why would this be?

(7) Supp Fig 3h-k - "PRP4K RNAi in eye discs resulted in reduced phosphorylation of Yki and YkiS168 but had no effect on Yki2SA or Yki3SA (Supplementary Fig. 3h-k)." - This is not convincing compared to the control track "-" - should quantify this to make it clearer.

(8) Line 143 - "To determine whether PRP4K directly phosphorylates Yki" - it would be helpful to the reader if it is described in the results that PRP4K was purified from bacterially expressed protein, to avoid any questions about co-purifying protein kinases from eukaryotic sources.

(9) Line 184 - Fig 5A and 5B - should state in the results or figure legend what dataset was used for this analysis.

(10) Line 196 - Fig 5f - "Cell growth" implies an increase in cell size - the methods state that cell numbers were determined - if this is the case relabel as "cell proliferation"
"Invasion" is not an accurate term when referring to a scratch wound assay that measures physiological epithelial sheet migration - should describe as "cell migration". Was mitomycin C added to prevent proliferation in this expt (it was not mentioned in the methods)? If not, how can it be concluded that this is just a migration effect?
It would be interesting to examine 3D cultures in matrigel for the effect of PRP4K on invasive properties.

(11) Graphs Fig 3i, h, and 4k need the Y-axis labelled, and Graphs Fig 5i, n - need the units on the

X-axis added.

(12) Since PRP4K phosphorylates similar sites on Yki/YAP to Wts/Lats some discussion on how PRP4K is related to the Wts/Lats family of kinases would be helpful to the reader. It seems like PRP4K is of the DUAL SPECIFICITY TYROSINE-PHOSPHORYLATION REGULATED KINASES - and is autoactivated by autophosphorylation on a tyrosine residue - is it thought that PRP4K is constitutively active, or is there anything known about how it is regulated?

Minor comments:

(1) It would help the reader if there were subheadings in the text summarizing the major points

(2) Some of the genetic nomenclature needs tightening up - genes should be referred to as they are listed in flybase - eg *hpo*, *yki*, *sd* should all be lowercase, it should be *FRT80A* and *FRT82B* and all genetic elements should be in italics.

(3) There are some typos that need correcting - eg "ey-Fly" should be "eyFLP", "batman" should be "bantam", "rabbit anti-Lat1" should be "rabbit anti-Lats1"

Reviewer #3 (Remarks to the Author):

In this manuscript Cho and collaborators show that nuclear protein kinase PRP4K has a crucial role in determining the phosphorylation status and sub-cellular distribution of both *Drosophila* Yki and human Yap/Taz factors of the Hippo pathway.

They also show that PRP4K is important to mediate the effect of starvation on the Hippo pathway and to restrict Yki-driven tissue overgrowth in *Drosophila*. Finally they map the Yki/Yap/Taz sites phosphorylated by PRP4K.

Their data are consistent with a model whereby "PRP4K plays an evolutionary conserved role in modulating the HIPPO pathway activity", "providing a second line of defence to prevent aberrant nuclear accumulation and activation of Yki/Yap".

The manuscript is well written and easy to follow. Experiments are carefully controlled and performed. Conclusions are supported by the results and are of general interest.

Minor points

1) It would be nice to have a scheme of the HIPPO pathway and cytoplasmic kinases at the beginning of the manuscript to help readers that are not familiar with this subject

2) Figure 4. Legend to panel c) says that "PRPK4 excludes Yap nuclear localisation" and "Hek293 cells were stained with Yki antibodies". Is this correct?

3) In the first and third lane of Figure 4 panel c (GFP and PRP4KRk respectively) YAP appears to be completely nuclear. However in Figure 4 panel e and in Figure 3 panel h one has the impression that the protein is always preferentially cytoplasmic. Moreover the protein is still 20% when all the Serine have been mutated, suggesting that additional mechanisms may operate. It would be nice if the authors could better comment the data.

Reviewers' comments:

Reviewer #1 (Remarks to the Author):

In this study, Cho et al discovered that PRP4K, a protein kinase that has been implicated in splicing, regulates Yki/Yap activity. In Drosophila, knockdown of PRP4K enhances the Yki overexpression phenotypes, both tissue growth and target gene expression, while overexpression of PRP4K suppresses the Yki overexpression phenotypes. PRP4K is found to act downstream or parallel to Hpo. Wts inhibits Yki by direct phosphorylation, including phosphorylation of S168 that creates 14-3-3 binding and induces Yki cytoplasmic localization. Interestingly, PRP4K also directly phosphorylates Yki on the Wts consensus sites S111 and S259, but not on the primary Wts site S168. Through an unknown mechanism, PRP4K phosphorylation increases Yki cytoplasmic localization, therefore inhibiting Yki activity. Evidence is presented to show that Wts and PRP4K phosphorylate Yki in the cytoplasm and nucleus, respectively. Further studies in mammalian cells suggest a similarly conserved function and mechanism of PRP4K in YAP and TAZ regulation.

This study provides interesting data to support a model that PRP4K can directly phosphorylate Yki/Yap on the Lats consensus sites, yet the PRP4K phosphorylation sites are not entirely overlapping with Wts/LATS. Besides the conventional Hippo pathway components, many genes have been implicated in Yap regulation in a "Hippo independent" manner, yet some of the reported new "regulators" do not stand further scrutiny. Available evidence overwhelmingly supports a model that Wts/LATS are probably the most important regulators of Yki/Yap. To appreciate a functional significance of PRP4K in Yki regulation, more evidence is needed to compare the relative contribution of PRP4K and Wts in Yki inhibition.

The phenotypes of PRP4K mutation and Wts mutation in tissue growth, Yki cellular localization and phosphorylation, and target gene expression should be compared side-by-side. This is very important, so the readers can appreciate the true physiological importance of PRP4K in Yki regulation.

Based on the proposed mechanism that Wts and PRP4K phosphorylate Yki on different sites in different subcellular compartment, double knockdown of Wts and PRP4K should be performed. The model predicts that double knockdown should produce additive or synergistic effects on Yki localization and tissue growth.

Response

I completely agree with the reviewer that Wts/LATS are the most important regulators of the Yki/Yap in the Hpo pathway. Our data are also consistent with Wts/LATS being the major kinase for Yki/Yap whereas PRP4K is a minor one that acts as a backup mechanism because inactivation of Wts posterior to the morphogenetic furrow (GMR>Wts-RNAi) resulted in enlarged eyes (revised Fig. 1n) whereas inactivation of PRP4K (GMR>PRP4K-RNAi) did not cause a discernable phenotype (revised Fig. S1b-d). In fact, the eye overgrowth phenotype caused by Wts RNAi was so strong that double knockdown of Wts and PRP4K did not further enhance the phenotype (our unpublished observation). In addition, Wts RNAi in wing discs resulted in more robust Yki nuclear localization (see Fig. 5 in Li et al., Cell Discovery 2015) than PRP4K RNAi (Fig. 2). Furthermore, we compared Yki mobility shift on phos-tag gel, which is a readout of Yki phosphorylation, in PRP4 RNAi, Wts RNAi, or Wts and PRP4 double RNAi eye imaginal discs. We found that Wts RNAi resulted a more dramatic reduction of Yki phosphorylation compared with PRP4 RNAi and that double RNAi further reduced Yki phosphorylation (revised Fig. S3n). Based on these results, we included several statements in the text to emphasize the predominant role of Wts over PRP4K:

---"Wts contributes more toward phosphorylating Yki than PRP4K"--- (p7, the end of paragraph one)

---"However, under physiological condition where the core Hpo kinase cassette is intact, Wts/Lats1/2 appears to play a more dominant role in phosphorylating and inhibiting Yki/Yap/Taz, which explains why loss of Wts but not PRP4K resulted in dramatic tissue overgrowth"--- (p12, the end of paragraph one)

Fig.3f. The data is very clean. However, using very short peptides as kinase substrates could produce misleading information. Is it possible to repeat the experiments with full length Yki and show that S111 and S250 are the primary PRP4K sites?

Response

In the experiments shown in Fig. S3f and Fig. S3j, mutating S111 and S250 in full-length Yki abolished PRP4K-induced mobility shift of the Yki variant (Yki2SA), suggesting that PRP4K phosphorylates S111 and S250 in the context of full-length Yki. In most cases, the specificity of kinase phosphorylation sites is determined by short consensus sequences (for example, RRXXS for a PKA site), which is not influenced by the neighboring sequence at a distance. We believe that the Yki sequences included in the GST-Yki fusion proteins we used for PRP4K substrates contain enough sequences flanking the phosphorylation sites and thus unlikely caused unspecific phosphorylation. In

addition, the consensus site we obtained for PRP4K explains why YkiS168 and YapS127 cannot be phosphorylated by PRP4K because they contain Pro at +2 position whereas PRP4K prefers an acidic residue at this position. Remarkably, substitution of P with D (S168PD) conferred phosphorylation of mutated site by PRP4K (Fig. 3f).

Fig. S3h-i. The data is rather strange and inconsistent with data in Fig.3h. Fig.3h shows that YkiS168A is much more nuclear than Yki2SA that is almost identical to Yki wild type, suggesting S168 being the primary regulatory site in Yki. However, in Fig. S3h-l, patterns on phostag gel of YkiS168A is almost identical to wild type Yki while Yki2SA (Fig.S3j) abolishes the mobility shift, suggesting that S168 has little impact on Yki phosphorylation. How these data can be reconciled?

Response

For unknown reason, S168 site phosphorylation did not seem to cause mobility shift of Yki on the phos-tag gel while phosphorylation of at S111 and S250 contributed to most of the observed mobility shift; therefore, the mobility shift of Yki variant does not reflect the status of Yki phosphorylation at S168. Using a phospho-S168 specific antibody, we found that neither loss of PRP4K nor its overexpression affected S168 phosphorylation (Fig. S3l, m). Since YkiS168 phosphorylation plays a major role in the regulation of Yki nuclear localization because it confers 14-3-3 binding, and since Wts but not PRP4K can phosphorylate S168, this may explain why loss of Wts has more profound effect on Yki activity than loss of PRP4K.

Fig.4d-e. The starvation-induced YAP mobility shift and localization should be compared with LATS1/LATS2 knockout cells to show the relative contribution of LATS and PRP4K in YAP regulation.

Response

We examined starvation-induced YAP mobility shift in PRP4K RNAi, LATS1/2 KO, and PRP4K RNAi + LATS1/2 KO background and found that either LATS1/2 or PRP4K inactivation resulted in a dramatic reduction whereas combined LATS1/2 and PRP4K inactivation completely abolished starvation-induced YAP mobility shift. These data have been incorporated into revised Fig. 4f. Consistent with this finding, a previous study showed that inactivation of LATS1/2 blocked starvation-induced YAP nuclear exclusion (Meng et al., Nature Communication 2015), similar to what we found for PRP4K inactivation (revised Fig. 4e). Hence, both PRP4K and LATS1/2 are critical for starvation-induced Yap phosphorylation; however, this is not a physiological setting because YAP was mainly localized in the nuclei before the starvation (revised Fig. 4e). We believe that under physiological conditions where Yki/Yap is mainly localized in the cytoplasm, Wts/LATA1/2 is the major kinase for Yki/Yap.

Reviewer #2 (Remarks to the Author):

This paper by Jiang and colleagues presents evidence for a new member of the Hippo pathway, the kinase PRP4K, that acts by phosphorylating Yki/YAP thereby promoting its nuclear export. The study is novel and significant as it relates to a new regular of the important Hippo tissue growth control pathway. PRP4K has been previously found to play roles in RNA slicing and the spindle assembly checkpoint, and more recently in EGFR membrane trafficking (Corkery et al., 2017) - so it appears to have pleiotropic functions - some discussion of these other roles and how the current role in the Hippo pathway might relate to these other findings would be helpful to the reader.

The study is well written and the data mostly clear, however it should be strengthened by quantification, particularly in instances where the effects seen are small.

Specific comments:

(1) It is now standard practise that effects observed on eye sizes should be quantified. This applies to Fig 1, Fig Supp 1 and Fig 3J.

Response

We have provided quantification of eye size (see revised Fig1e, k, p-q; Fig. 3j'; Fig. S1e, j, o-p)

(2) Figure 1D "PRP4K overexpression suppressed diap1-GFP3.5 expression and eye overgrowth induced by GMR>Yki" - This is not convincing - needs quantification of posterior vs anterior staining (where the GMR driver is not expressed) is needed for this and the other eye disc images.

Response

We have replaced with better images for diap-GFP staining in Fig. 1a"-d". The quantification of *diap1-GFP3.5* expression posterior vs anterior staining has been provided in revised Fig 1f.

(3) PRP4K- EY11156 mutant - This is an enhancer P line - where is it localised in the gene and how is it orientated? Some explanation is needed for this allele in the methods.

Response

We provided a description of this P element insertion line in the method: "EY11156 is a lethal mutation caused by an insertion of *P{EPgy2}* at the 5' region of the *CG7028/PRP4K* coding sequence. *P{EPgy2}* is in the opposite orientation relative to *CG7028* transcription".

(4) Figure 1f - There seem to be different shades of red/orange - perhaps due to some heterozygous tissue remaining? To illustrate the growth advantage more clearly could also examine this in the larval eye disc to see if there is a clonal advantage of PRP4K at this stage, as is expected for Hippo pathway mutations.

Response

The red tissues are heterozygous whereas the orange ones are homozygous for *prp4k* mutation. The *prp4k* mutant tissues (orange tissue in revised Fig. 1h) occupy a larger proportion of the eyes than the wild type control clones (white tissue in revised Fig. 1e), suggesting a growth advantage associated with *Prp4k* mutant tissues, allowing them to compete more favorably with the heterozygous wild type tissue and contribute more to adult eyes. Cell competition in eye development provides a more sensitive assay for even a subtle change in cell growth. We have examined *prp4k* mutant clones at larval stage in both wing and eye discs and did not observe an obvious increase in clone size compared with control clones (our unpublished observations). This is likely due to: 1) relatively mild defect in Hpo signaling and 2) the involvement of *Prp4K* in other cellular processes such as splicing and cytokinesis. It is conceivable that the defects outside the Hpo pathway may cause growth disadvantage that could off-set the growth advantage caused by gain of *Yki* activity.

(5) Supp Fig 2e-i - "Coexpression of PRP4K but not PRP4KKR, or knockdown of Yki suppressed the elevated ex-lacZ expression caused by hh>PRP4Ki" - Is this statement correct? Yki knockdown does look like it suppresses ex-lacZ upregulation? To make this analysis more convincing could quantify the staining in the posterior versus anterior wing domains. It should be stated in the sup fig leg what the Ci staining detects.

Response

The reviewer is correct that knockdown of *Yki* did suppress elevated *ex-lacZ* expression. We meant that "knockdown of *Yki* suppressed the elevated *ex-lacZ* expression caused by *hh>PRP4Ki*". We have now rephrased the sentence to make it clearer. We also stated in the legend that *Ci* staining marks the anterior compartment where *hh-Gal4* is not expressed.

(6) Supp Fig 3d - Seems like there is less overall Yki in the LMB Hpo lane - ie would expect that if Yki can't exit the nucleus Hpo-Wts phosphorylation of it would decrease, but there should be more unphosphorylated Yki (lower band)? Is there really less Yki here? If so why would this be?

Response

There is no evidence that *Yki* level is regulated by Hpo-Wts phosphorylation in *Drosophila*. The slightly reduced overall level could be due to unequal loading. In fact, the lower band (unphosphorylated form of *Yki*) is slightly increased in LMB treated lane compared with untreated lane so that the ratio of shifted signal vs un-shifted signal is reduced after LMB treatment in Hpo overexpressing cells, suggesting that Hpo-mediated phosphorylation was compromised when *Yki* nuclear export was inhibited.

(7) Supp Fig 3h-k - "PRP4K RNAi in eye discs resulted in reduced phosphorylation of Yki and YkiS168 but had no effect on Yki2SA or Yki3SA (Supplementary Fig. 3h-k)." - This is not convincing compared to the control track "-" - should quantify this to make it clearer.

Response

We agree that the effect of *PRP4K* RNAi on the mobility shift of *Myc-Yki* or *Myc-YkiS168* was subtle. The reduction of phosphorylation is mainly reflected by a downshift of the top bands. We have now included a bracket to highlight the degree of mobility shift (revised FigS3h-i).

(8) Line 143 - "To determine whether PRP4K directly phosphorylates Yki" - it would be helpful to the reader if it is described in the results that PRP4K was purified from bacterially expressed protein, to avoid any questions about co-purifying protein kinases from eukaryotic sources.

Response

PRP4K is a large kinase (907 aa) and would be difficult to express and purified from bacteria. We used immunopurified PRP4K and its kinase-dead form PRP4K^{KR} as a control. Because immune-purified PRP4K^{KR} complex had no kinase activity, it is unlikely that the in vitro kinase activity associated with immunopurified PRP4K was due to other associated kinases. The same strategy has been used by other labs in the field to demonstrate that Wts directly phosphorylates Yki or MAP4K directly phosphorylate Lats (for example see Meng et al., Nature communication 2015). Nevertheless, we have removed the word "directly" to be more conservative.

(9) Line 184 - Fig 5A and 5B - should state in the results or figure legend what dataset was used for this analysis.

Response

We have now indicated in the figure legend that Data were obtained from KMPLoTdatabase (<http://kmplot.com/>).

*(10) Line 196 - Fig 5f - "Cell growth" implies an increase in cell size - the methods state that cell numbers were determined - if this is the case relabel as "cell proliferation"
"Invasion" is not an accurate term when referring to a scratch wound assay that measures physiological epithelial sheet migration - should describe as "cell migration". Was mitomycin C added to prevent proliferation in this expt (it was not mentioned in the methods)? If not, how can it be concluded that this is just a migration effect?
It would be interesting to examine 3D cultures in matrigel for the effect of PRP4K on invasive properties.*

Response

-We changed "cell growth" to "cell proliferation" accordingly.
-In the wound healing assay, we used low serum medium (1%FBS) with 100% cell confluence, which is a standard for this type of experiments. Cells did not proliferate under this condition.
-We have now included a transwell assay, a more standard assay for cell invasion, to show that PRP4K knockdown increased breast cancer cell invasion. These new data have been included in revised Fig. 5j.

(11) Graphs Fig 3i, h, and 4k need the Y-axis labelled, and Graphs Fig 5i, n - need the units on the X-axis added.

Response

We added the missing labels the above panels. Thanks.

(12) Since PRP4K phosphorylates similar sites on Yki/YAP to Wts/Lats some discussion on how PRP4K is related to the Wts/Lats family of kinases would be helpful to the reader. It seems like PRP4K is of the DUAL SPECIFICITY TYROSINE-PHOSPHORYLATION REGULATED KINASES - and is autoactivated by autophosphorylation on a tyrosine residue - is it thought that PRP4K is constitutively active, or is there anything known about how it is regulated?

Response

PRP4K is a poorly studied kinase in terms of biochemistry and how its kinase activity is regulated has remained unknown to our knowledge. Although PRP4K is annotated as a dual specificity kinase in the Flybase, there is no evidence that PRP4K is regulated by tyrosine phosphorylation or auto-phosphorylation.

Minor comments:

(1) It would help the reader if there were subheadings in the text summarizing the major points

We added subheadings. Thanks.

(2) Some of the genetic nomenclature needs tightening up - genes should be referred to as they are listed in flybase - eg hpo, yki, sd should all be lowercase, it should be FRT80A and FRT82B and all genetic elements should in italics.

We have changed FRT80 to FRT80A and FRT82 to FRT82B, and italicized all the genetic elements. In general, we use low case for genes and mutants and upper case for proteins and transgenes gene.

(3) There are some typos that need correcting - eg "ey-Fly" should be "eyFLP", "batman" should be "bantam", "rabbit anti-Lats1" should be "rabbit anti-Lats1"

We made the corrections. Thanks.

Reviewer #3 (Remarks to the Author):

In this manuscript Cho and collaborators show that nuclear protein kinase PRP4K has a crucial role in determining the phosphorylation status and sub-cellular distribution of both Drosophila Yki and human Yap/Taz factors of the Hippo pathway.

They also show that PRP4K is important to mediate the effect of starvation on the Hippo pathway and to restrict Yki-driven tissue overgrowth in Drosophila. Finally, they map the Yki/Yap/Taz sites phosphorylated by PRP4K. Their data are consistent with a model whereby "PRP4K plays an evolutionary conserved role in modulating the HIPPO pathway activity", "providing a second line of defense to prevent aberrant nuclear accumulation and activation of Yki/Yap".

The manuscript is well written and easy to follow. Experiments are carefully controlled and performed. Conclusions are supported by the results and are of general interest.

Minor points

1) It would be nice to have a scheme of the HIPPO pathway and cytoplasmic kinases at the beginning of the manuscript to help readers that are not familiar with this subject

Response

We thank the reviewer for the suggestion. In the revision, we included an introduction section to provide background information for the Hippo pathway.

2) Figure 4. Legend to panel c) says that "PRPK4 excludes Yap nuclear localisation" and "Hek293 cells were stained with Yki antibodies". Is this correct?

It should be Yap antibody. We made the correction. Thanks.

3) In the first and third lane of Figure 4 panel c (GFP and PRP4KRk respectively) YAP appears to be completely nuclear. However in Figure 4 panel e and in Figure 3 panel h one has the impression that the protein is always preferentially cytoplasmic. Moreover the protein is still 20% when all the Serine have been mutated, suggesting that additional mechanisms may operate. It would be nice if the authors could better comment the data.

The difference in the Yap nuclear localization between Fig4c and Fig. 4f is most likely due to a difference in cell density in these cultures. It has been shown that increased cell density promotes Yap cytoplasmic localization. Cells in Fig4c were cultured at relatively lower density than in Fig4e, which resulted in a more distinct Yap nuclear localization.

We have previously shown that Yki nuclear localization depends on Sd (Zhang et al., Dev Cell 2008). In the experiments shown in Fig. 3g-h, Myc-Yki and its variants were cotransfected with HA-Sd into S2 cells, Yki nuclear localization was variable among different cells, likely due variable ratio of Myc-Yki/HA-Sd in different cells. Even in the case of Myc-Yki3SA, some cells may have low HA-Sd relative to Myc-Yki3SA, leading to more Myc-Yki3SA in cytoplasm. In addition, Yki protein appears to have many predicted nuclear export signals and treating cells with LMB caused increased the nuclear localization of Myc-Yki3SA (Ren et al., Dev Biology 2010), suggesting that Yki is actively transported out of the nucleus.

Reviewer #1 (Remarks to the Author):

OK with the authors' responses. No further comment.

Reviewer #2 (Remarks to the Author):

In this revision of their manuscript, the authors have added new data, supplied quantification to relevant images and clarified the text. I am happy with the revisions the authors have made to the MS, and believe that it is now suitable to be accepted for publication in Nature Communications.

REVIEWERS' COMMENTS:

Reviewer #1 (Remarks to the Author):

OK with the authors' responses. No further comment.

Reviewer #2 (Remarks to the Author):

In this revision of their manuscript, the authors have added new data, supplied quantification to relevant images and clarified the text. I am happy with the revisions the authors have made to the MS, and believe that it is now suitable to be accepted for publication in Nature Communications.

Response

The reviewers did not have any concerns to be addressed